# Hydrogen bonds as molecular timers for slow inactivation in voltage-gated potassium channels

Stephan A Pless[1,2], Jason D Galpin[1,3], Ana P Niciforovic[1], Harley T Kurata[1], Christopher A Ahern[1,2,3]*

[1]Department of Anesthesiology, Pharmacology and Therapeutics, University of British Columbia, Vancouver, Canada; [2]Department of Cellular and Physiological Sciences, University of British Columbia, Vancouver, Canada; [3]Department of Molecular Physiology and Biophysics, University of Iowa, Iowa City, United States

**Abstract** Voltage-gated potassium (Kv) channels enable potassium efflux and membrane repolarization in excitable tissues. Many Kv channels undergo a progressive loss of ion conductance in the presence of a prolonged voltage stimulus, termed slow inactivation, but the atomic determinants that regulate the kinetics of this process remain obscure. Using a combination of synthetic amino acid analogs and concatenated channel subunits we establish two H-bonds near the extracellular surface of the channel that endow Kv channels with a mechanism to time the entry into slow inactivation: an intra-subunit H-bond between Asp447 and Trp434 and an inter-subunit H-bond connecting Tyr445 to Thr439. Breaking of either interaction triggers slow inactivation by means of a local disruption in the selectivity filter, while severing the Tyr445–Thr439 H-bond is likely to communicate this conformational change to the adjacent subunit(s).

*For correspondence:
christopher-ahern@uiowa.edu

Competing interests: The authors declare that no competing interests exist.

## Introduction

Enzymes and catalytic proteins have evolved to balance the thermodynamic challenges of stability and substrate throughput (*Shoichet et al., 1995*). Ion channels, for instance, must efficiently interconvert between open, closed and inactivated states to regulate ionic flux across biological membranes. Even small alterations in their function that change the rates of isomerization between states can underlie inherited or acquired diseases (*Hille, 2001*). Voltage-gated potassium (Kv) channels are tetrameric membrane proteins with a central ion-conducting pore domain, surrounded by four voltage-sensor domains (VSDs), which tightly regulate the conductive state of the pore domain (*Figure 1A*). The pore domain consists of two transmembrane helices (S5–S6) connected by a re-entrant pore helix, which forms the selectivity filter. Kv channels negatively regulate conductance after channel opening through a process termed inactivation. The rate and extent of inactivation exhibit considerable isoform-dependent differences, which are reflected in the physiological contributions of these channels to cellular excitability in neuronal and cardiac tissues (*Bean, 2007*; *Smith et al., 1996*; *Spector et al., 1996*; *Sanguinetti and Tristani-Firouzi, 2006*; *Aldrich et al., 1979*). Inactivation can be described by two kinetically and mechanistically distinct processes termed fast (or 'N-type') inactivation and slow (or 'C-type') inactivation (*Hoshi et al., 1991*; *Kurata and Fedida, 2006*; *Hoshi and Armstrong, 2013*). While the former results from a channel peptide docking within the open cytoplasmic entrance to the permeation pathway (*Hoshi et al., 1990*, *1991*; *Zhou et al., 2001*), the latter is assumed to involve highly cooperative local conformational changes near the selectivity filter, a notion supported by electro-physiological, structural and computational approaches (*Lopez-Barneo et al., 1993*; *Yellen et al., 1994*; *Liu et al., 1996*, *1997*; *Starkus et al., 1997*; *Kiss et al., 1999*; *Cordero-Morales et al., 2006*,

**eLife digest** Proteins are made from long chains of smaller molecules, called amino acids. These chains twist and bend into complex three-dimensional shapes, and sometimes two or more chains, or 'subunits', are packed into a protein. These shapes are often held together by hydrogen bonds between some of the amino acids. Moreover, since the shape of a protein defines its function, some proteins must be able to switch between different shapes to function properly.

Ion channels are proteins that form pores through cell membranes, allowing ions to flow in and out of the cell. Potassium ion channels, which are found in neurons and heart muscle cells, have four subunits that move to open or close the central pore in response to various signals.

The closing of the channels can be 'fast' or 'slow'. When the channels are closed quickly (called fast inactivation), a small part of the protein 'plugs' the pore from the inside of the cell. However, the mechanism behind slow inactivation remained obscure. It was thought to involve hydrogen bonds between some of the bulky amino acids that are found at the edge the pore. However, testing this hypothesis—by replacing these amino acids with alternatives that cannot form hydrogen bonds—was tricky because none of the 20 naturally occurring amino acids were alike enough to be suitable replacements.

Now, Pless et al. have overcome this limitation by using synthetic amino acids that form hydrogen bonds that are stronger or weaker than those formed by the amino acids they are replacing. The results suggest that two types of hydrogen bond keep the pore open: one is a bond between two amino acids in the same subunit, and the other is an inter-subunit bond between amino acids in neighbouring subunits. Pless et al. suggest that opening the channel causes small movements that gradually weaken, and eventually break, these bonds in one of the four subunits. Specific amino acids within the pore are then free to twist and—via a cascade of similar movements in the other three subunits—block the pore and halt the flow of ions. As such, these networks of hydrogen bonds act as pre-set breaking points allowing channels to close, even in response to continued stimulation.

Since regulated potassium channel activity underpins healthy neurons and heart muscles; understanding what controls their inactivation rate may lead to new approaches to tune their activity and treatments for important diseases.

*2007*; *Peng et al., 2007*; *Cuello et al., 2010a, b*; *Cordero-Morales et al., 2011*; *Ostmeyer et al., 2013*; *van der Cruijsen et al., 2013*). However, and despite this available data, fundamental questions persist regarding the precise molecular determinants that mediate the rate of slow inactivation, the basis for cooperativity and the relationship between slow inactivation and the structural integrity of the selectivity filter (*Hoshi and Armstrong, 2013*). An ongoing challenge is to elucidate how these proteins precisely time the entry into the slow inactivated state in the presence of a sustained voltage stimulus.

The Kv channel pore domain contains intermeshed aromatic side chains, an arrangement termed the 'aromatic cuff' that is located at the extracellular end of the selectivity filter and pore helix (*Figure 1*). This highly conserved region has long been suggested to play a part in the stability of the pore and likely slow inactivation (*Doyle et al., 1998*; *Larsson and Elinder, 2000*; *Kurata and Fedida, 2006*), yet the dynamic rearrangements during inactivation have remained poorly resolved given that many side chains within this region are intolerant to replacement, likely due to the large chemical and steric changes produced by standard mutagenesis. Here, we bypass this experimental hurdle by employing subtle synthetic derivatives of naturally occurring side chains in combination with concatenated subunits to probe the inter- and intra-subunit atomic determinants that control the onset and cooperativity of slow inactivation in Kv channels.

## Results

### Defining an intra-subunit H-bond between Asp447 and Trp434

Crystallographic data demonstrate the close physical proximity of highly conserved Asp and Trp side chains within the same subunit of potassium channels (*Figure 2A*) (*Doyle et al., 1998*; *Long et al.,*

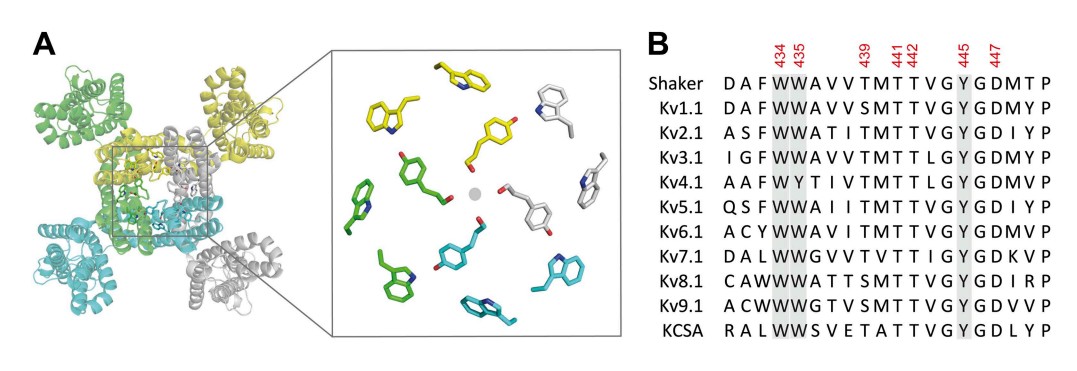

**Figure 1**. The aromatic cuff is part of a highly conserved region in potassium channels. (**A**) Top view of a Kv channel based on the structure of the tetrameric Kv1.2/2.1 chimera (PDB 2R9R; individual subunits are colored in gray, cyan, green and yellow, respectively). The inset shows a magnified view of the side chains that form the 'aromatic cuff': Trp434, Trp435 and Tyr445 (by numbering in Shaker potassium channels). Note the backbone carbonyls are shown for Tyr445 to highlight their role in the coordination of potassium ions (gray circle); (**B**) Sequence alignment of the pore helix and the selectivity filter of various potassium channels: Shaker (GI:288442), Kv1.1 (GI:119395748), Kv2.1 (GI:84570020), Kv3.1 (GI:298603), Kv4.1 (GI:8272404), Kv5.1 (GI:24418476), Kv6.1 (GI:24418479), Kv7.1 (GI:6166005), Kv8.1 (GI:7657289), Kv9.1 (GI:219520418), and KCSA (GI: 61226909). Side chains constituting the aromatic cuff are highlighted in gray (see above) and all positions studied here are indicated using their numbering in Shaker potassium channels (these residues correspond to Trp362, Trp363, Ser367, Thr369, Thr370, Tyr373 and Asp375 in the Kv1.2/Kv2.1 [voltage-gated potassium channel isoforms 1.2 and 2.1] chimera crystal structure [PDB 2R9R]).

2005). Mutations at these positions (Asp447 and Trp434 by Shaker numbering, which is used throughout the manuscript) elicit drastic changes in channel function, including effects on inactivation (*Perozo et al., 1993*; *Molina et al., 1997*, *1998*; *Yang et al., 1997*; *Loots and Isacoff, 2000*; *Loboda et al., 2001*; *Cordero-Morales et al., 2011*). However, functional studies have not demonstrated the chemical basis of the interaction between these two side chains. One previously proposed (*Cordero-Morales et al., 2011*; *Hoshi and Armstrong, 2013*), but untested possibility is that a H-bond is formed between the hydrogen on the indole nitrogen of Trp434 and the carboxylate moiety of Asp447, and disrupting this H-bond would promote conformational changes associated with slow inactivation. If correct, then a side chain such as Glu (with altered stereochemistry, but a conserved negatively charged carboxylate), might weaken the interaction but not abolish it completely. By contrast, the virtually isosteric but uncharged Asn side chain should not form a significant H-bond with Trp434 and should thus result in dramatically accelerated inactivation. Indeed, the Asp447Glu mutation leads to a rapid and complete decrease in ionic current (*Molina et al., 1997*, *1998*), and this loss in conductance can be drastically slowed by addition of extracellular tetraethyl ammonium (TEA). This deceleration of current decay in the presence of extracellular TEA is a hallmark of slow inactivation and can serve to discriminate this process from other phenotypes, such as fast inactivation or generic protein dysfunction (*Grissmer and Cahalan, 1989*; *Choi et al., 1991*; *Molina et al., 1997*). The charge-neutralizing Asp447Asn mutation resulted in gating currents only, generally interpreted to reflect an inactivation phenotype too rapid to resolve (*Figure 2B* and *Table 1*) (*Hurst et al., 1996*; *Yang et al., 1997*; *Loots and Isacoff, 1998*). Conversely, manipulations of the putative H-bonding partner Trp 434 should cause complimentary effects. For example, if the hydrogen on the indole nitrogen of Trp434 does, indeed, play a critical role in slow inactivation, then removing it should drastically increase the rate of inactivation. However, there are no naturally occurring Trp derivatives that can faithfully test this hypothesis without perturbing the local structure. Other (smaller) aromatic side chains markedly alter channel function: Tyr in position 434 speeds up slow inactivation (and is sensitive to TEA, *Figure 2—figure supplement 1*), while Phe results in gating currents only (*Figure 2C*; *Table 1*) (*Perozo et al., 1993*; *Yang et al., 1997*; *Cordero-Morales et al., 2011*). Thus, while structural data suggests a possible H-bond between Trp434 and Asp447, the available functional data cannot definitely discriminate between the roles of side chain size and/or volume or hydrogen bonding ability being the major determinant of slow inactivation at position 434. We therefore employed synthetic derivatives of Trp to test the hypothesis that a H-bond between Trp434 and Asp447 is a rate-controlling interaction for slow inactivation. If this H-bond is a

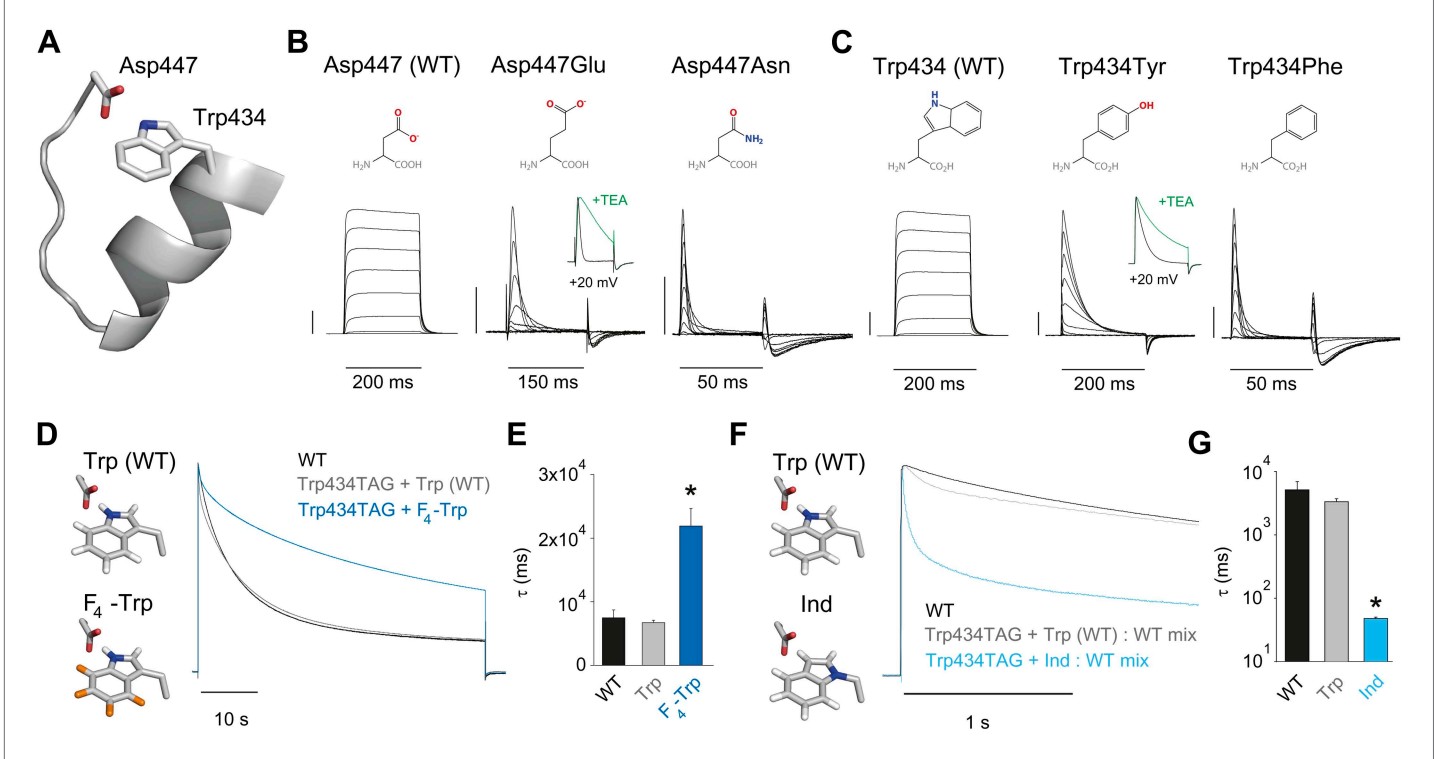

**Figure 2**. Evidence for an intra-subunit H-bond between Asp447 and Trp434. (**A**) Structure of a Kv1.2/2.1 chimera (2R9R) pore region demonstrating the physical proximity of Asp447 and Trp434 (Shaker residue numbering); (**B**) Chemical structures of side chains at position 447 and (normalized) representative currents for Asp447 (WT), Asp447Glu and Asp447Asn (−80 mV to +20 mV in 10 mV increments). Inset for Asp447Glu shows the traces recorded for a pulse to +20 mV in the absence and presence of 30 mM TEA (see **Figure 2—figure supplement 1** for further details); (**C**) Chemical structures of side chains at position 434 and (normalized) representative currents for Trp434 (WT), Trp434Tyr and Trp434Phe (−80 mV to +20 mV, 10 mV increments). Inset for Trp434Tyr shows the traces recorded for a pulse to +20 mV in the absence and presence of 30 mM TEA (see **Figure 2—figure supplement 1** for further details). Unlike the Trp434Phe mutant the Asp447Asn mutant remained nonconducting on the Thr449Val background as shown in **Figure 2—figure supplement 2**; the vertical scale bar indicates 1 µA; (**D**)/(**F**) Left panels: model for the Asp447–Trp434 pair (based on PDB 2R9R) for either Trp and F4-Trp (**D**) or Trp and Ind (**F**); Right panels: normalized sample currents for WT, Trp434TAG + Trp or Trp434TAG + F4-Trp (**D**) or WT, Trp434TAG + Trp: WT or Trp434TAG + Ind: WT (**F**). The initial faster decay of the gray traces in (**D**) and (**F**) may indicate that a very small fraction of channels has incorporated amino acids other than the one ligated to the tRNA, see text for details. Note that for the experiments in (**F**) and (**G**) WT cRNA was mixed with Trp434TAG cRNA to account for the fact the Trp434TAG + Ind alone did not yield measurable ionic currents (see 'Materials and methods' for details). Note the different time scales in (**D**) and (**F**); (**E**)/(**G**) Averaged inactivation time constants for the constructs shown in (**D**) and (**E**), respectively. Single exponentials were fit to the entire (45 s) depolarizations in (**D**), while in (**F**) the time constants for Ind were determined by fitting only the initial 500 ms of the depolarization (5 s for WT and Trp434TAG + Trp: WT); Note the logarithmic scaling in (**G**); *p<0.05 (WT vs mutants).

The following figure supplements are available for figure 2:

**Figure supplement 1**. Tuning the Asp447–Trp434 intra-subunit H-bond.

**Figure supplement 2**. Unlike Trp434Phe, Asp447Asn remains nonconducting on the Thr449Val background.

critical determinant of the rate of slow inactivation, predicted outcomes are that strengthening the interaction should decelerate inactivation, while weakening the putative H-bond would be expected to increase the rate of inactivation. To this end, we introduced $F_4$-Trp (**Figure 2D**), a fluorinated Trp derivative that increases the acidity of the hydrogen on the indole nitrogen (**Deutsch and Taylor, 1987**, **1989**), while leaving side chain size and hydrophobicity intact. This slowed the rate of slow inactivation by threefold (**Figure 2E**). However, this slowing is preceded by an initial faster inactivating component in the Trp434TAG + Trp trace (gray trace in **Figure 2D**), which may stem from a small degree of nonspecific incorporation of (endogenous) amino acids other than the one ligated to the tRNA (in this case Trp). This likely produces an underestimate of the true slowing of inactivation through fluorination as all naturally occurring amino acids other than Trp in position 434 lead to accelerated slow

**Table 1.** Mutants that result in gating currents only show very similar gating charge-voltage (QV) relations

| Construct | V1/2 (mV) | Z | n |
|---|---|---|---|
| Trp434Phe | −52.9 ± 0.9 | 5.2 ± 0.3 | 6 |
| Asp447Asn | −53.4 ± 0.9 | 5.3 ± 0.3 | 6 |
| Thr439Val | −51.0 ± 1.4 | 5.5 ± 0.3 | 5 |
| Tyr445Ala | −50.3 ± 2.1 | 5.3 ± 0.4 | 4 |

Displayed are the values for the midpoints ($V_{1/2}$), the amount of gating charge (Z) of the QVs (derived from the OFF gating currents) and the number of experiments conducted (n).

inactivation. As a chemical complement, we sought to incorporate 2-amino-3-indol-1-yl-propionic acid (Ind, *Figure 1F*) (*Lacroix et al., 2012*), a synthetic isosteric Trp derivative in which the indole hydrogen cannot act as a H-bond donor. As co-injection of Trp434TAG cRNA and Ind-coupled tRNA alone did not yield measurable ionic currents, Wild type cRNA was mixed with Trp434TAG cRNA and Ind-coupled tRNA. However, even in the presence of WT (Wild type) subunits, the resulting heteromeric channel population displayed a 70-fold increase in inactivation rate (*Figure 2G*), demonstrating that breaking the proposed Asp447–Trp434 H-bond in the absence of steric perturbation leads to substantially accelerated inactivation. Similar to our recordings with Trp434TAG + Trp alone, we observed an initial fast inactivating component with the Trp434TAG + Trp: WT mix (gray trace in *Figure 2F*), possibly indicating a small degree of nonspecific incorporation of endogenous amino acids in position 434. However, this also is unlikely to have a major impact on our results because, first, the resulting fast component is a minor component only visually discernible during the initial phase (less than 1 s) of the depolarization and, second, the overall measured effect is likely more affected (slowed) by the presence of the WT subunits that were necessary to obtain ionic currents with Ind-containing subunits. We thus conclude that slow inactivation is amenable to the simple atomic 'push/pull' of a single H-bond, and manipulation of the strength of this H-bond generates a spectrum of inactivation rates, that can be tuned to occur faster or slower than what is observed in WT channels.

Slow inactivation is highly cooperative (*Ogielska et al., 1995*; *Panyi et al., 1995*; *Yang et al., 1997*) and the single mutation, Trp434Phe, in the first of four concatenated subunits accelerates slow inactivation (*Yang et al., 1997*). We reasoned that the phenotype obtained by breaking the putative H-bond through manipulations at Trp434 should be mimicked by mutations at the complementary Asp447 site. Indeed, the Asp447Glu mutation in the first of four concatenated *Shaker* subunits accelerated inactivation rates, similar to those obtained from Trp434Phe concatemers (*Figure 3A,B*). Together, these data support the notion that the strength of the Asp447–Trp434 intra-subunit H-bond is directly correlated with slow inactivation rate, suggesting that breaking of this interaction is an intrinsic timing mechanism that tightly regulates Kv channel activity.

## A novel inter-subunit H-bond with a pivotal role in slow inactivation

Structural evidence suggests that Trp435 (*Figure 4A*) forms an inter-subunit H-bond via its hydrogen on the indole nitrogen with the Tyr445 hydroxyl (*Doyle et al., 1998*; *Larsson and Elinder, 2000*; *Kurata and Fedida, 2006*), and therefore substitution of Tyr or Phe for Trp435 would be expected to disrupt this H-bond, and potentially accelerate inactivation (as observed for aromatic substitutions of the adjacent Trp434 residue). However, while the Trp435Ala mutation produced non-functional channels (as suggested by the absence of ionic or gating currents), Tyr and Phe substitutions at position 435 resulted in WT-like slow inactivation rates (*Figure 4B,C*), ruling out a role for Trp435 H-bonding in slow inactivation. However, the Tyr445Phe mutation results in a mix of gating current and ionic current, with markedly accelerated slow inactivation (*Harris et al., 1998*) (a phenotype antagonized by TEA) (*Figure 4D*, *Figure 4—figure supplement 1*). Furthermore, Tyr445Ala channels exhibited gating currents akin to Trp434Phe channels (*Figure 4D*; *Table 1*) (*Heginbotham et al., 1994*). Interestingly, crystallographic data (*Doyle et al., 1998*; *Long et al., 2007*) place the Tyr445 hydroxyl within 3 Å of the hydroxyl moiety of a conserved Thr or Ser side chain (Thr439 in *Shaker*, *Figure 1B*), raising the intriguing possibility of an uncharacterized inter-subunit interaction between Tyr445 and Thr439. Consistent with this possibility, the Thr439Val mutant exhibited exclusively gating currents (*Figure 4E*), while the Thr439Ser mutation resulted in a modest threefold faster inactivation

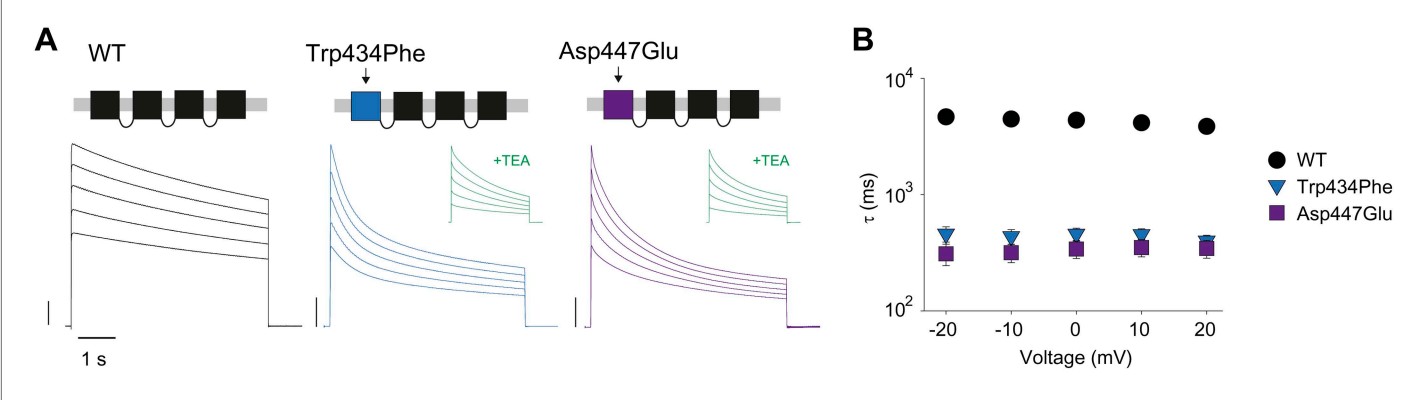

**Figure 3**. Concatemers support the notion of an intra-subunit H-bond between Asp447 and Trp434. (**A**) Concatemer structure and (normalized) representative currents (5 s pulses from −20 mV to +20 mV, 10 mV increments) for WT, Trp434Phe and Asp447Glu concatemers, respectively. The vertical scale bars indicate 2 µA. The insets show recording from the same cells in the presence of 30 mM TEA; (**B**) Averaged inactivation rates (logarithmic scaling) over different voltages for the constructs shown in (**A**). Note that for the Trp434Phe and Asp447Glu concatemers only the first 2 s of the depolarization were fit with a single exponential. To avoid a potential bias of this approach, we have also analyzed the time to half-maximal current for all constructs. Importantly, this approach yielded similar results, see *Figure 3—figure supplement 1*.

The following figure supplements are available for figure 3:

**Figure supplement 1**. Comparison of different metrics to determine the rate of inactivation.

than observed for WT channels, likely indicating a minor role for the Thr439 methyl group in slow inactivation (*Figure 4E*, *Figure 4—figure supplement 3*). Further, and unlike the Trp434Phe mutation (*Kitaguchi et al., 2004*), introducing Tyr445Ala, Tyr445Val or Thr439Val on the Thr449Val background did not slow inactivation to an extent that ionic currents could be observed (*Figure 4—figure supplement 3*). While these data point to an inter-subunit H-bond between Tyr445 and Thr439, they do not inform on the cooperativity between individual subunits or whether this interaction contributes to the same extent as the Asp447–Trp434 H-bond. However, we speculated that breaking this inter-subunit H-bond may have a more pronounced effect when introduced in only a single subunit, compared to the intra-subunit Trp434–Asp447 H-bond. Consistent with this possibility, either the Tyr445Ala or the Thr439Val mutation in the first of four concatenated *Shaker* subunits (*Figure 5A*) had similar phenotypes, with a clearly biphasic inactivation phenotype composed of fast (around 50 ms) and WT-like slow (around 3 s) components (*Figure 5B*). The fast component was affected by TEA, implicating a slow inactivation mechanism (*Figure 5—figure supplement 1*). The sizable gating currents at hyperpolarized potentials (*Figure 5—figure supplement 2*) suggest that either mutation (one per concatenated tetramer) reduces the ratio of ionic current to gating charge at a given voltage, an effect that would arise if a significant portion of channels rapidly adopt a non-conducting conformation. To further test this possibility, the pore blocker agitoxin II (*Eriksson and Roux, 2002*; *Banerjee et al., 2013*) was used to assay the gating currents as a metric for normalization of the number of channels present in the cell, and thus permitting an estimate of the relative reduction in ionic current in the mutant concatemers relative to WT concatemers. Indeed, we found the ratio of ionic current to gating charge to be significantly reduced in both mutant concatemers (*Figure 5C*), suggesting that a sizable proportion of channels rapidly enter an inactivated state upon depolarization. This behavior is further illustrated in *Figure 5D*, where currents from Tyr445Ala or Thr439Val concatemers were normalized to WT (by gating charge), thus emphasizing the very rapid and near-complete inactivation in Tyr445Ala and Thr439Val concatemers. These experiments establish a previously unidentified inter-subunit H-bond between Thr439 and Tyr445 that controls slow inactivation in Kv channels.

## Thr441 and Thr442 are essential for channel function but not slow inactivation

Thr441 and Thr442 are highly conserved amongst Kv channels and are favorably located at the junction of selectivity filter and pore helix (*Figure 6A*) for a possible role in pore stability and/or slow inactivation. We aimed to compare the relative contribution of Thr441 and Thr442 to slow inactivation with more

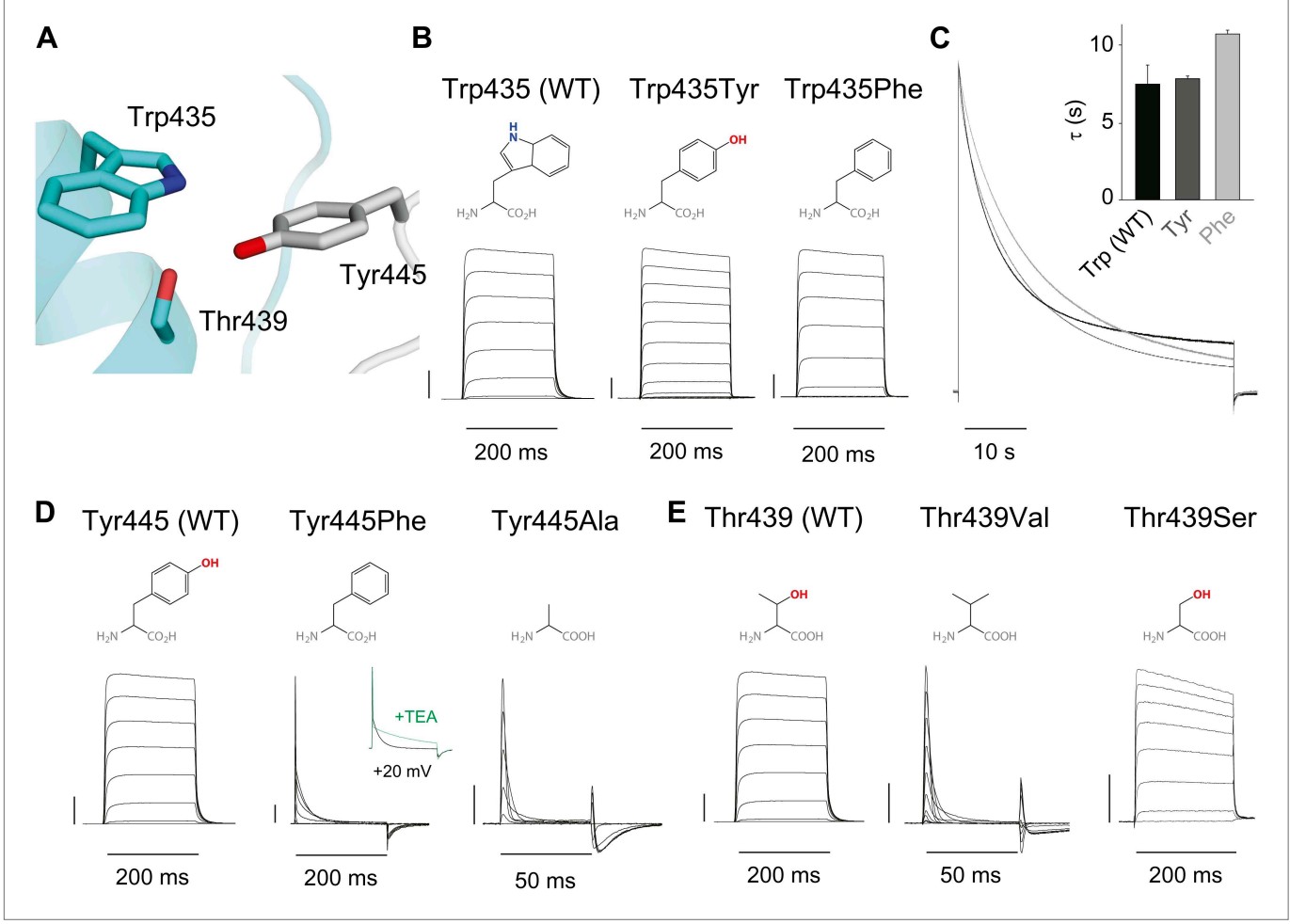

**Figure 4**. An inter-subunit H-bond connects Tyr445 with Thr439, not Trp435. (**A**) Structure of a Kv1.2/2.1 chimera (2R9R) pore region demonstrating the physical proximity of Tyr445 to both Thr439 and Trp435 on the adjacent subunit (Shaker residue numbering). Note that the position equivalent to position 439 in Shaker (Thr439) is a serine in the Kv1.2/2.1 chimera; (**B**) Chemical structures of side chains at position 435 and representative currents for Trp435 (WT), Trp435Tyr and Trp435Phe (−80 mV to +20 mV, 10 mV increments). The vertical scale bar indicates 1 µA; (**C**) Representative normalized currents for a 45 s depolarization to +20 mV for Trp435 (WT), Trp435Tyr and Trp435Phe. Inset shows average inactivation time constants for the constructs shown in (**B**) (single exponential fit over the entire duration of the depolarization); (**D**)/(**E**) Chemical structures and (normalized) representative currents for different side chains in position 445 (**D**) and 439 (**E**), respectively (−80 mV to +20 mV, 10 mV increments). The vertical scale bars indicate 1 µA, note the different time scales. Inset for Tyr445Phe shows the traces recorded for a pulse to +20 mV in the absence and presence of 30 mM TEA (see *Figure 4—figure supplement 1* for further details). The Tyr445Ala and Thr439Val mutants remained nonconducting on the Thr449Val background as shown in *Figure 4—figure supplement 2*. The inactivation time constant (τ) for Thr439Ser was 948 ± 30 ms compared to 3247 ± 186 ms for WT channels (see *Figure 4—figure supplement 3* for further details).

The following figure supplements are available for figure 4:

**Figure supplement 1**. Tuning the Tyr445–Thr439 inter-subunit H-bond.

**Figure supplement 2**. Tyr445Ala, Tyr445Val and Thr439Val remain nonconducting on the Thr449Val background.

**Figure supplement 3**. The methyl moiety of Thr439 may play a minor role in inactivation.

extracellular structural elements of the selectivity filter. Interestingly, mutations here produce vastly different outcomes (depending on the amino acid substitution), including loss-of-function, alterations in open state stability, and the appearance of subconductance states with diminished selectivity (*Yool and Schwarz, 1991*; *Heginbotham et al., 1994*; *Zheng and Sigworth, 1997*). Consistent with these reports, we found that valine substitutions a 441 and 442 had severe consequences: while Thr442Val

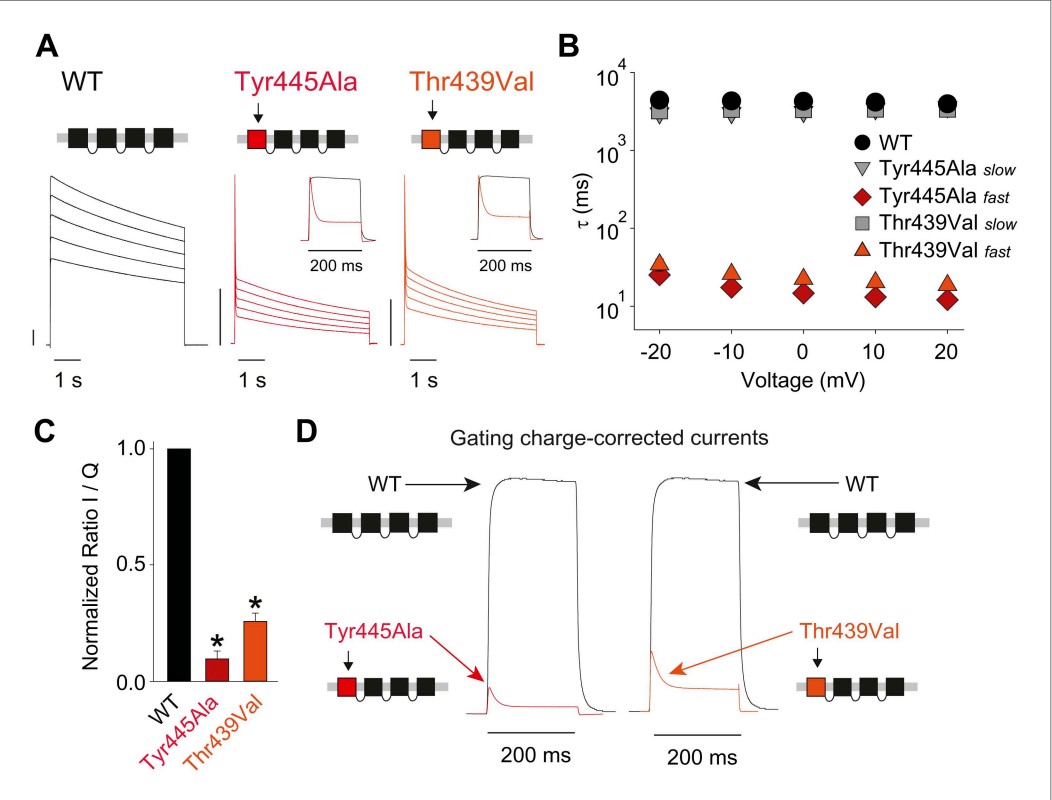

**Figure 5**. Breaking the Tyr445–Thr439 inter-subunit H-bond results in rapid inactivation. (**A**) Concatemer structure and representative currents (5 s pulses from −20 mV to +20 mV, 10 mV increments) for WT, Tyr445Ala and Thr439Val concatemers, respectively. The vertical scale bars indicate 2 μA. The insets compare normalized currents in response to a +20 mV step for WT and Tyr445Ala or Thr439Val concatemers over a short (200 ms) time scale (see ***Figure 5—figure supplement 1*** for details on TEA sensitivity; see ***Figure 5—figure supplement 2*** for gating currents at hyperpolarized potentials with Tyr445Ala and Thr439Val concatemers); (**B**) Averaged inactivation time constants over different voltages for the constructs shown in (**A**), with rates for Tyr445Ala and Thr439Val concatemers split into fast and slow components (single exponentials were fit to the first 50 ms and the remainder of the depolarization, respectively). Note the logarithmic scaling; (**C**) Ratio of maximal ionic current to gating charge (both recorded at +20 mV) for WT, the Tyr445Ala and the Thr439Val concatemers (gating currents were recorded in the presence of 10 μM agitoxin II, not shown); *p<0.05 (WT vs mutants); (**D**) Comparison of ionic currents recorded at +20 mV normalized to the amount of gating charge recorded from WT concatemers and the Tyr445Ala (left panel) and the Thr439Val (right panel) concatemers, respectively.

The following figure supplements are available for figure 5:

**Figure supplement 1**. Characterizing the Thr439Val and Tyr445Ala concatemers.

**Figure supplement 2**. Pronounced gating currents at hyperpolarized potentials in Tyr445Ala and Thr439 Val concatemers.

---

displayed a non-expressing phenotype (***Figure 6C***) (***Zheng and Sigworth, 1997***), the Thr441Val mutation resulted in voltage-dependent currents both in the inward and the outward direction, as well as reduced potassium selectivity (***Figure 6B***, ***Figure 6—figure supplement 1***), suggesting a significant perturbation of the local structure. However, Thr441Ser channels displayed a WT-like GV whereas Thr442Ser channels displayed a modest left-shifted activation relationship, yet both had inactivation behaviors similar to WT channels (***Figure 6B,C***, ***Figure 6—figure supplement 2***), supporting the notion that the hydroxyl moieties at positions 441 and 442 support normal pore function (in addition to the proposed role of the Thr442 methyl group in ion binding (***Rossi et al., 2013***)). However, despite the disruptive phenotypes observed in homotetrameric Thr441Val or Thr442Val channels, single subunit mutations had minimal effects in the background of a concatenated tetramer (***Figure 6D,E***).

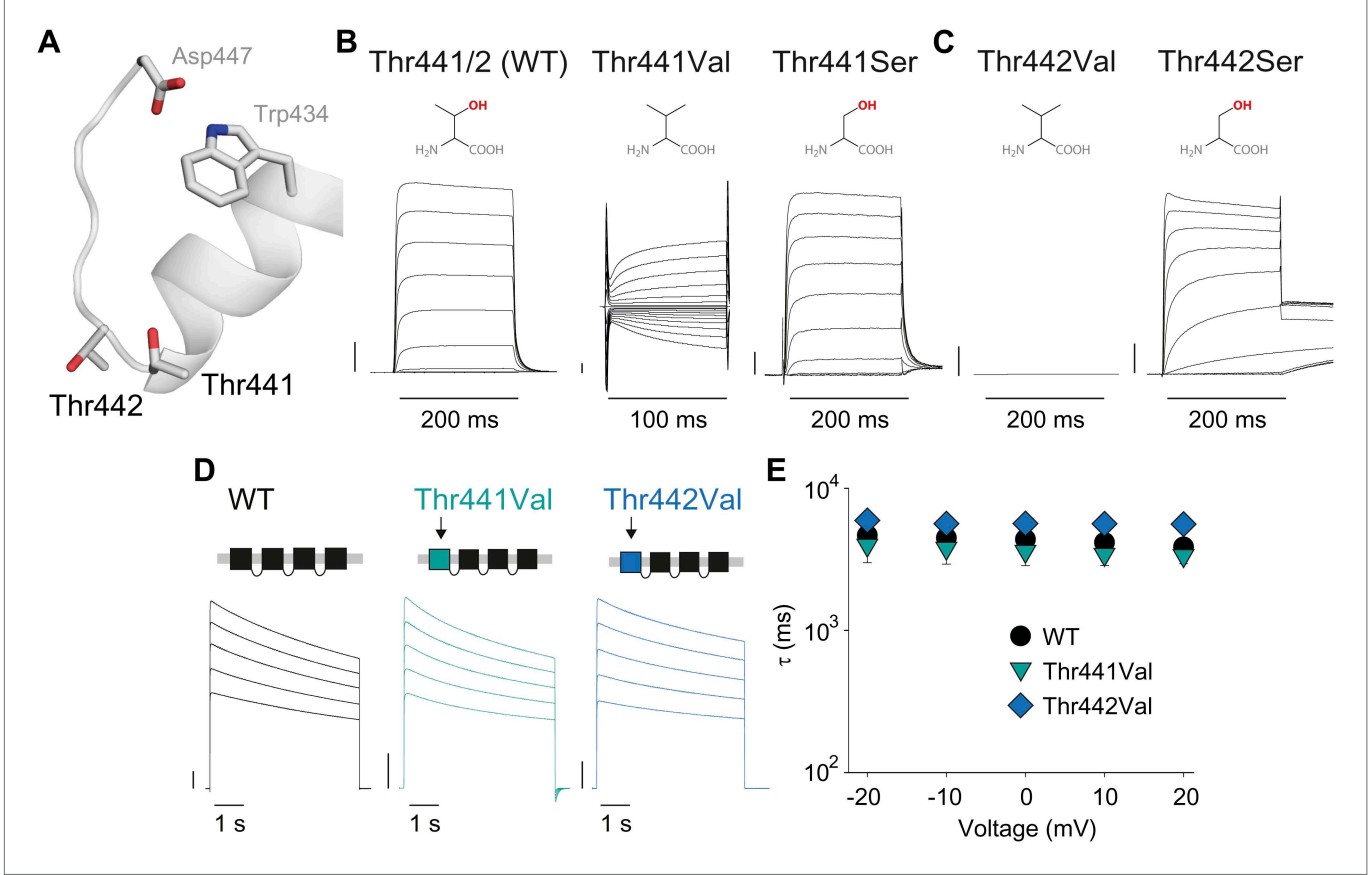

**Figure 6**. Thr441 and Thr442 are critical to channel function but not inactivation. (**A**) Structure of a Kv1.2/2.1 chimera (2R9R) pore region highlighting the positions of Thr441 and Thr442 at the bottom of the selectivity filter (Asp447 and Trp434 are shown for reference; all by Shaker numbering); (**B**, **C**) Chemical structures and (normalized) representative currents for different side chains in position 441 (**B**) and 442 (**C**), respectively (−80 mV to +20 mV, 10 mV increments for WT, Thr441Ser and Thr442Ser; −200 mV to +150 mV, 10 mV increments for Thr441Val and a single pulse to +20 mV for Thr442Val). Note that recordings for Thr441Val were conducted from a holding potential of 0 mV with no leak subtraction. The vertical scale bars indicate 1 μA. See ***Figure 6—figure supplement 1*** for details on the loss of potassium selectivity of Thr441Val. GVs and inactivation behavior of Thr441Ser and Thr442Ser are shown in ***Figure 6—figure supplement 2***; (**D**) Concatemer structure and representative currents (5 s pulses from −20 mV to +20 mV, 10 mV increments) for WT, Thr441Val and Thr442Val concatemers, respectively. The vertical scale bars indicate 2 μA. The small inward tail current for Thr441Val could indicate reduced potassium selectivity; (**E**) Averaged inactivation time constants over different voltages for the constructs shown in (**D**); similar results were obtained with longer (20 s) depolarizations, see ***Figure 6—figure supplement 3***.

The following figure supplements are available for figure 6:

**Figure supplement 1**. The Thr441Val mutation results in a loss of potassium selectivity.

**Figure supplement 2**. Hydroxyl moieties are critical to channel function in positions 441 and 442.

**Figure supplement 3**. Comparison of inactivation time constants using different pulse durations.

Thus, hydroxyl removal at either 441 or 442 produces channel phenotypes that are very mild compared to perturbations within the aromatic cuff, and importantly, their effects are not propagated to other subunits in the channel tetramer, suggesting they are unrelated to the cooperative mechanism of slow inactivation. We believe this is an important finding as it demonstrates that severe functional consequences of mutations at the selectivity filter are not necessarily linked to changes in slow inactivation.

## Discussion

Despite intense experimental scrutiny for almost 25 years, the molecular and atomic origin(s) of the ability of Kv channels to enter a non-conducting conformation in the presence of a sustained (voltage)

stimulus has remained enigmatic. A major challenge with addressing the contribution of individual side chains in the selectivity filter and pore helix to slow inactivation is that many amino acids lack naturally occurring analogs that allow subtle manipulation without dramatic disruption of the overall structure of this critical protein region. Furthermore, since slow inactivation is tightly coupled to ion occupancy in the selectivity filter, it has been difficult to distinguish direct effects on the mechanistic underpinnings of slow inactivation from indirect effects arising from changes in the structural integrity of the selectivity filter. Here, we overcome this hurdle by employing subtle synthetic analogs of naturally occurring amino acids and by introducing isolated mutations in single subunits.

When using concatenated *Shaker* constructs to introduce single mutations in a fourfold symmetric channel, it is crucial to confirm that the concatenated subunits do not form functional channels that vary in their stoichiometry from that predicted from the cloning strategy. Albeit possible (*McCormack et al., 1992*; *Hurst et al., 1995*), we believe the constructs used here assemble correctly for three reasons. First, the Thr439Val concatemers and the Tyr445Ala concatemers showed almost complete inactivation over a period of only 200 ms (when corrected for the amount of gating charge). Second, we observed very low ratios of $I_{max}$ to $Q_{max}$ for Thr439Val concatemers and Tyr445Ala concatemers. Both scenarios are not compatible with the idea of a significant WT-only channel subpopulation; Lastly, Sigworth and co-workers have used the same concatemers and successfully demonstrated that channels containing only a single mutated subunit generally assemble in the correct stoichiometry (*Yang et al., 1997*). We conclude that the (vast majority of) concatemers assemble correctly, although we cannot ultimately rule out a small subpopulation of channels with WT-like slow inactivation.

Further, we employed fluorinated Trp derivatives, which have been used extensively to probe electrostatic (cation-pi) interactions (*Dougherty, 1996*) between Trp side chains and organic cations as fluorination allows a step-wise dispersion of the electronegative surface potential of aromatic side chains (*Pless and Ahern, 2013*). As such, our finding that $F_4$-Trp in position 434 significantly slows channel inactivation could be interpreted as a result of a cation-pi interaction at Trp434 that is being diminished by fluorination. However, if this were true, Ind, a synthetic amino acid which lacks H-bonding ability, should have no effect on channel inactivation as it is isosteric and isoelectric to the native Trp side chain. By contrast, we observe a substantial increase in the rate of slow inactivation with Ind in position 434, a result not compatible with the notion of an energetically significant cation-pi interaction at Trp434. We thus conclude that it is the ability of the indole nitrogen to participate in a H-bond that regulates the strength of the intra-subunit interaction between Trp434 and Asp447.

Together, our experimental approaches provide strong evidence for two H-bonds that are critical for slow inactivation of Kv channels: one that confers stability within an individual subunit (the Trp434–Asp447 interaction), and a second that stabilizes the relative orientation of two adjacent subunits (the Tyr445–Thr439 interaction) (*Figure 7*). Although disruption of the Trp434–Asp447 interaction has

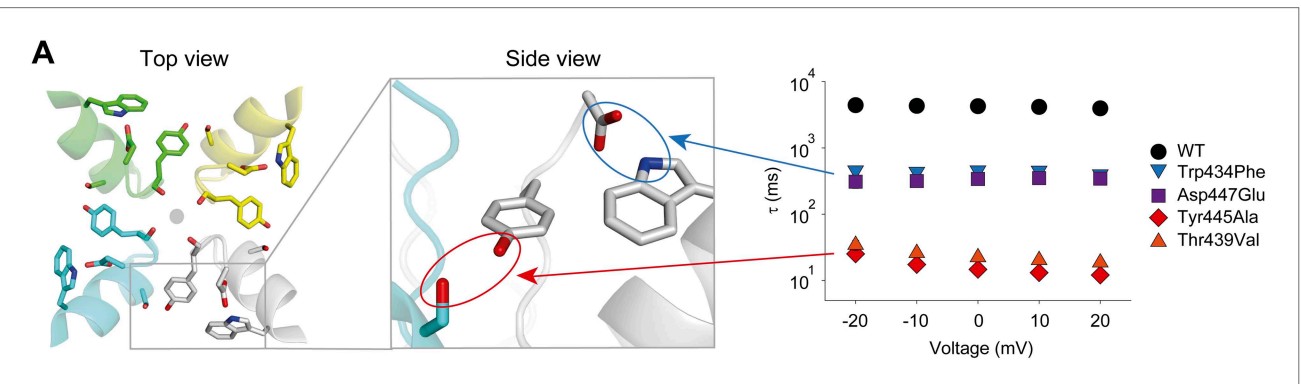

**Figure 7**. A network of inter- and intra-subunit H-bonds regulates slow inactivation. (**A**) The left panel shows a top view of pore helix and selectivity filter (based on the Kv1.2/2.1 chimera structure (2R9R); individual subunits are colored in gray, cyan, green and yellow, respectively). Note the backbone carbonyls are shown for Tyr445 to highlight their role in the coordination of potassium ions (gray circle). The center panel highlights the two proposed H-bonds: Thr439–Tyr445 (inter-subunit, red oval,) and Asp447–Trp434 (intra-subunit, blue oval), all by Shaker numbering. The panel on the right compares the averaged inactivation time constants over a range of voltages for different concatemers (data reproduced from *Figure 3* and *Figure 5*; note that for Tyr445Ala and Thr439Val only the fast components are displayed). Note the arrows pointing to the respective interactions in the model in the center.

profound effects on slow inactivation, breaking of the Tyr445–Thr439 interaction elicits more functionally significant phenotypes. We surmise that these comparatively more severe phenotypes seen when disrupting the Tyr445–Thr439 pair arise from its location at the inter-subunit interface, but cannot exclude the possibility that this difference arises from the fact that the Tyr445 backbone carbonyl is also directly involved in coordinating permeant ions. Furthermore, the Tyr445–Thr439 interaction is, to our knowledge, the first evidence for an inter-subunit interaction contributing to slow inactivation, possibly providing an explanation for the observed subunit cooperativity during slow inactivation. However, although previous studies have suggested evidence for both constriction (*Baukrowitz and Yellen, 1996*; *Liu et al., 1996*, *1997*) and dilation (*Hoshi and Armstrong, 2013*) of the selectivity filter, the data here is not definitive in distinguishing these models of slow inactivation.

The notion that side chains critical to slow inactivation cluster around the 'aromatic cuff' (formed between the extracellular end of the selectivity filter and the pore helix) is further supported by the marked differences between side chains at the outer vs the inner end of the selectivity filter and pore helix: only those located around the 'aromatic cuff' result in notable effects on slow inactivation that are propagated to the entire channel (*Figures 2–5*), while those residing in the middle or lower section of the selectivity filter do not affect slow inactivation (see *Figure 6* for positions 441 and 442; see (*Heginbotham et al., 1994*) for position 443).

Overall, the results point towards an intriguing molecular explanation for the mechanism of slow inactivation: upon depolarization and channel opening, the stability of the channel open state is proportional to the strength of two H-bonds that regulate entry into slow inactivation, thus endowing Kv channels with an intrinsic timing mechanism that tightly regulates their biological activity. During a sustained voltage stimulus, channels experience a sequential breaking of the Trp434–Asp447 and Tyr445–Thr439 H-bonds and given the relative arrangement of their hydroxyl moieties this would likely result in an anti-clockwise swivel movement of the Tyr445 backbone carbonyl away from the permeation pathway, ultimately disrupting the coordination and occupancy of potassium ions at the outer end of the selectivity filter. Such a scenario would lead to mutual repulsion between the Tyr445 backbone carbonyls of the remaining three subunits (*Almers and Armstrong, 1980*; *Hoshi and Armstrong, 2013*), further lowering filter-occupancy at its outer mouth. The resulting strain could trigger a cascade of disrupted H-bonds critical to inactivation near the extracellular end of the selectivity filter in all subunits, ultimately resulting in a fully inactivated channel.

## Materials and methods

### Molecular biology and in vivo nonsense suppression

*Shaker* IR (Inactivation Removed by deletion of amino acids 6–46) cDNA in pBSTA was used as the parent clone unless stated otherwise (note that Cys301 and Cys308 were present). For mutations, standard site-directed mutagenesis was employed in combination with automated sequencing to confirm successful incorporation of mutations. For experiments with concatenated *Shaker* tetramers (*Yang et al., 1997*), mutations were introduced in the first of the four subunits only: the first subunit was subcloned into pBSTA with SacI and XbaI, and standard site-directed mutagenesis was used to introduce mutations followed by sequence verification. Next, the mutated construct was subcloned (with SacI and XbaI) back into the parent concatemer.

For electrophysiology experiments, Stage V-VI *Xenopus* oocytes were prepared, and injected with cRNA transcribed with the T7 mMessage mMachine kit (Ambion, Austin, TX) as previously described (*Pless et al., 2013*). Oocytes were incubated at 18°C and all recordings were conducted within 12–72 hr after injection. The fluorinated Trp derivative $F_4$-Trp (4,5,6,7-$F_4$-Trp) was purchased from Asis Chem (Watertown, MA) and the Trp analog 2-Amino-3-indol-1-yl-propionic acid (Ind) was synthesized as described previously (*Lacroix et al., 2012*). The principle of the in vivo nonsense suppression methodology is outlined elsewhere (*Pless and Ahern, 2013*). In short, nitroveratryloxycarbonyl (NVOC) was used to protect the amine of the synthetic amino acid, while the carboxyl group was activated as the cyanomethyl ester for coupling to the dinucleotide pdCpA (Dharmacon, Lafayette, CO). The resulting product was stored in DMSO at −80°C before enzymatic ligation to a modified (G73) *Tetrahymena thermophila* tRNA, which was synthesized using an oligonucleotide by Integrated DNA Technologies (Coralville, IA) as a template. The NVOC protection group of the aminoacylated tRNA-UAA was removed directly prior to co-injection with the cRNA by UV irradiation for 8 min at 400 W. In a typical experiment, 10–80 ng of tRNA-UAA and 25–50 ng of cRNA were co-injected in a 50 nl vol. In control experiments,

the tRNA coupled to pdCpA (without an appended synthetic amino acid) was co-injected with the Trp434TAG cRNA. The control did not yield currents larger than for uninjected oocytes, ruling out significant levels of non-specific amino acid incorporation, or re-charging of the tRNA with endogenous amino acids. Note that incorporation of Ind at position 434 did not result in measurable ionic currents. This was expected given the results with the conventional Trp434Tyr and Trp434Phe mutants, as side chains in position 434 with no propensity to contribute to a H-bond (such as Ind) were shown to result in gating currents only. However, the nonsense suppression method is generally not efficient enough to establish current levels of necessary magnitude to resolve gating currents, even for sites with exceptionally high incorporation efficiency (*Pless et al., 2011, 2013*). We thus co-injected WT *Shaker* cRNA with the Trp434TAG cRNA for incorporation of Ind. Despite the high degree of cooperativity between subunits (*Figure 3* and *Figure 5*), this likely results in slower and less complete slow inactivation than would be expected for Ind-containing subunits alone and the 70-fold increase in inactivation rate (*Figure 2*) is likely to be an underestimate of the real acceleration of inactivation induced by the Ind side chain in position 434.

### Electrophysiology and data analysis

Two electrode voltage-clamp recordings were conducted with an OC-725C voltage clamp (Warner, Hamden, CT) in standard Ringers solution (in mM): 116 NaCl, 2 KCl, 1 MgCl$_2$, 0.5 CaCl$_2$, 5 HEPES (pH 7.4). TEA (St. Louis, MO) and agitoxin-II (Alomone Labs, Jerusalem, Israel) were dissolved in Ringers and stored at −20°C until use. Glass microelectrodes with resistances of 0.1–1 MΩ were back-filled with 3 M KCl. Currents were acquired using leak subtraction and from a holding potential of −80 mV, unless stated otherwise. To obtain conductance-voltage (GV) relationships, isochronal tail current amplitudes were plotted vs the depolarizing pulse potential. All data = mean ± SEM; Student's *t* test was used to determine statistically significant differences. To obtain gating charge-voltage (QV) relationships, the total area of the off gating charge was plotting against the depolarizing pulse potential.

## Acknowledgements

We would like to thank Dr Fred Sigworth for kindly providing the concatenated *Shaker* WT clone.

## Additional information

### Funding

| Funder | Grant reference number | Author |
|---|---|---|
| National Institutes of Health | GM106569 | Christopher A Ahern |
| Canadian Institutes of Health Research | MOP-56858 | Harley T Kurata |

The funders had no role in study design, data collection and interpretation, or the decision to submit the work for publication.

### Author contributions

SAP, Conception and design, Acquisition of data, Analysis and interpretation of data, Drafting or revising the article; JDG, APN, Acquisition of data, Contributed unpublished essential data or reagents; HTK, CAA, Conception and design, Analysis and interpretation of data, Drafting or revising the article

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
