## [Decision Letter]

Thank you for sending your work entitled “Hydrogen bonds as molecular timers for slow inactivation in voltage-gated potassium channels” for consideration at *eLife*. Your article has been favorably evaluated by a Senior editor, a Reviewing editor, and 2 reviewers.

The Reviewing editor and the two reviewers discussed their comments before we reached this decision, and the Reviewing editor has assembled the following comments to help you prepare a revised submission.

Using the atomic structure of Kv1.2/2.1 as the roadmap, Pless et al. ask a series of specific questions regarding the mechanism of C-type inactivation focusing on the hydrogen bonds near the selectivity filter. For example, is the (probable) hydrogen bond between W434 and D447 important in C-type inactivation of Shaker potassium channels? Despite the efforts by many, the exact mechanism of C-type inactivation remains mysterious and any specific and firm mechanistic information will be appreciated by those in the field and this study has potential to make an important contribution. The use of unusual amino acids using the in vivo nonsense suppression method brings a fresh approach to study C-type inactivation. The overall experimental rationale/design is straightforward, the experiments are elegant and the conclusions are for the most part solid.

There are two major central issues that must be addressed:

1) The overly simplistic treatment of inactivation kinetics must be corrected, taking into account the existence of non-exponential decay.

Perhaps the most problematic issue with the study described in this manuscript may lie in the data analysis/interpretation and the data description (e.g., the manuscript text). Many of the sample ionic currents involving the in vivo nonsense suppression method exhibit double-exponential inactivation kinetics (e.g., Figure 2 blue, Figure 2 gray, and Figure 2 W434F blue). Except for Figure 1 gray and blue, it is not clear what is responsible for the double exponential character. Is it possible that the nonsense suppression method was not perfect? Given the double exponential character, it is also unclear how the currents were analyzed to estimate the time constant values plotted in the figures. It is also uncertain how the “rate of entry into slow inactivation” often mentioned in the text is estimated. Additionally, the manuscript does not report any attempt to separate the entry and recovery rates – this would be important because many of the sample currents shown (e.g., Figure 2; Figure 3) suggest that the inactivated state at depolarized test voltages was not an absorbing state and the recovery rate constant value was appreciable (as evidenced by “steady-state” currents in the sample data shown). If the authors really wish to address the stability of the open state (as opposed to the stability of the inactivated state), they may wish to separate them out.

2) The work has little power (or relevance) in distinguishing between the pore collapse and pore dilation hypotheses of inactivation. Speculation along these lines should be eliminated. Likewise the distinction between pore stability and inactivation is weak and should be toned down.

The data presented on the perturbation analysis done on positions Thr441 and Thr442 although extremely interesting are not conclusive evidence to rule out “pore stability and selectivity filter from C-type inactivation”. These far-reaching conclusions have not been systematically addressed by the authors. The fact that the authors introduce the mutation on the first subunit of the tandem tetramer and they saw no effect implies that one subunit is not sufficient to affect slow inactivation and it might need more than one, but concluding that we should “conceptually dissociate pore stability and selectivity filter from C-type inactivation” is mere speculation that is not demonstrated in this paper.

---

## [Author Response]

*1) …Many of the sample ionic currents involving the* in vivo *nonsense suppression method exhibit double-exponential inactivation kinetics (e.g.,*
Figure 2
*blue,*
Figure 2
*gray, and*
Figure 2
*W434F blue). Except for*
Figure 1
*gray and blue, it is not clear what is responsible for the double exponential character. Is it possible that the nonsense suppression method was not perfect*?

The basis for the multi-exponential nature of slow inactivation is not known but this complication has not hampered the numerous previous studies that have nonetheless approximated the decay as a single exponential process. In this particular case, we believe the reviewers are referring to the small rapid components at the very beginning of the depolarizing pulses. The reviewers are most likely correct in attributing this to imperfect nonsense suppression. Indeed, we often suspect that a small degree of non-specific incorporation of amino acids other than the one ligated to the synthetic tRNA that we inject. Since Trp434 mutations generate extremely rapid inactivation (cooperatively), even a small amount of non-specific incorporation could be sufficient to translate into a rapid component in macroscopic currents. A small rapid component was observed irrespective of the ligated amino acid – the ‘WT rescue’ experiment that introduces Trp by nonsense suppression shows a similar rapid component, and so we think that this property is best attributed to some imperfections of the nonsense suppression. We were unsure of the importance of going into explicit detail about this in the original submission, so we thank both reviewers for highlighting this point.

We hope to emphasize that this issue does not significantly impact on the interpretation of our results: it is known from other studies as well as ours that 1) amino acids other than Trp induce a rapidly inactivating phenotype, and 2) that the process of slow inactivation is highly cooperative. Therefore, any ‘imperfections’ of the in vivo nonsense suppression method would lead to phenotypes that inactivate faster than WT. As we still observe a significant slowing in the rate of inactivation for F_4_-Trp, the values obtained by us are actually quite likely to serve as an underestimate of the real extent of slowing induced by fluorination of F_4_-Trp. Similarly, we cannot rule out that some of the fast inactivating component of the Ind-containing channels is ‘contaminated’ through non-specific incorporation of amino acids other than Ind. However, the fast component is very small compared to that observed with Ind. Moreover, Ind-containing subunits had to be mixed with WT subunits to obtain ionic currents (which will significantly slow inactivation). Therefore, values obtained through this approach also reflect acceleration induced through incorporation of Ind. We have now explicitly mentioned these limitations of our approach in the Results section, and we hope this will add significant candor and clarity to the text:

“However, this slowing is preceded by an initial faster inactivating component in the Trp434TAG + Trp trace (gray trace in Figure 2), which may stem from a small degree of nonspecific incorporation of (endogenous) amino acids other than the one ligated to the tRNA (in this case Trp)…”

*Given the double exponential character, it is also unclear how the currents were analyzed to estimate the time constant values plotted in the figures*.

We have now clarified this by including explicit information on how the data were fit in all figure legends.

Regarding the specific case of the Trp434Phe concatemers, we did observe varying rates of inactivation in both Trp434Phe and Asp447Glu concatemers, with some being well-fit by a single exponential, while others may have been better fit with multiple time constants. To account for this observation (and to facilitate comparison with other mutants) we had limited the analysis to the first 2 seconds of the depolarization (for the Trp434Phe and Asp447Glu concatemers), and this is now explicitly mentioned in the figure legend of Figure 3. Further, and to exclude any bias introduced by this approach, we also used in parallel a ‘time-to-half- maximal-current’ metric as a more generic quantification of inactivation and obtained virtually identical results (shown in Figure 3–figure supplement 3). Finally, we would like to again remark that the molecular basis for the occurrence of multiple time constants during inactivation does not only show up in our data set, but is a generally accepted phenomena that remains ill- defined, significantly complicating the interpretation of potential multi exponential decays.

*Additionally, the manuscript does not report any attempt to separate the entry and recovery rates – this would be important because many of the sample currents shown (e.g.,*
Figure 2*;*
Figure 3*) suggest that the inactivated state at depolarized test voltages was not an absorbing state and the recovery rate constant value was appreciable (as evidenced by “steady-state” currents in the sample data shown). If the authors really wish to address the stability of the open state (as opposed to the stability of the inactivated state), they may wish to separate them out*.

We appreciate this comment and realize the merits of separating the entry and recovery rates from inactivation. However, the recovery from inactivation in potassium channels has been recently shown to be a highly complex molecular orchestration that not only involves protein conformational changes, but is also primarily controlled by the water occupancy behind the selectivity filter (Ostmeyer et al., Nature, 2013). We thus believe that it would be beyond the scope of the current study to investigate effects on recovery from inactivation in a detailed manner. Furthermore, while our current data set does not allow to definitively exclude potential effects on recovery from slow inactivation, we believe major effects on recovery are unlikely for the following reason: in the simplified context of a two state model (open-inactivated), a faster recovery rate from inactivation would also be predicted to result in a faster initial decay that would result in an increased steady-state current (compared to WT). These criteria were not met by any of the mutants tested. Only Trp434TAG + F_4_-Trp displayed an increased steady-state current, but the concomitant initial acceleration of inactivation was due to the ‘imperfect’ incorporation of F_4_-Trp (as verified by the ‘imperfect’ incorporation of Trp; see also our above comments), making a major effect on recovery unlikely.

*2) The work has little power (or relevance) in distinguishing between the pore collapse and pore dilation hypotheses of inactivation. Speculation along these lines should be eliminated. Likewise the distinction between pore stability and inactivation is weak and should be toned down*.

*The data presented on the perturbation analysis done on positions Thr441 and Thr442 although extremely interesting are not conclusive evidence to rule out “pore stability and selectivity filter from C-type inactivation”. These far-reaching conclusions have not been systematically addressed by the authors. The fact that the authors introduce the mutation on the first subunit of the tandem tetramer and they saw no effect implies that one subunit is not sufficient to affect slow inactivation and it might need more than one, but concluding that we should “conceptually dissociate pore stability and selectivity filter from C-type inactivation” is mere speculation that is not demonstrated in this paper*.

We agree with the reviewers that our data has limited power to discriminate between these disparate concepts and in hindsight we feel we may have overstepped this interpretation in our data. Consequently, we have now removed all such conclusions regarding a potential discrimination between pore dilation vs collapse and pore stability vs slow inactivation. Specifically, we have made the following changes to the manuscript:

The final sentence of the Abstract has now been changed to: “...triggers slow inactivation by means of a local disruption in the selectivity filter...”

The second-to-last sentence of the Results section (“These data therefore serve to conceptually dissociate the notion of pore stability and selectivity filter from the mechanism of slow inactivation”) has been removed and the final sentence has been changed to:

“We believe this is an important finding as it demonstrates that severe functional consequences of mutations at the selectivity filter are not necessarily linked to changes in slow inactivation.”

Finally, we have now changed the final sentence of the second-to-last paragraph of the Discussion section to: “However, although previous studies have suggested evidence for both constriction and dilation of the selectivity filter, the data here is not definitive in distinguishing these models of slow inactivation.”